# Density Distribution Maps: A Novel Tool for Subcellular Distribution Analysis and Quantitative Biomedical Imaging

**DOI:** 10.3390/s21031009

**Published:** 2021-02-02

**Authors:** Ilaria De Santis, Michele Zanoni, Chiara Arienti, Alessandro Bevilacqua, Anna Tesei

**Affiliations:** 1Department of Medical and Surgical Sciences (DIMEC), Alma Mater Studiorum, University of Bologna, I-40138 Bologna, Italy; i.desantis@unibo.it; 2Interdepartmental Centre Alma Mater Research Institute on Global Challenges and Climate Change (Alma Climate), University of Bologna, I-40126 Bologna, Italy; 3Biosciences Laboratory, IRCCS Istituto Scientifico Romagnolo per lo Studio e la Cura dei Tumori (IRST) “Dino Amadori”, I-47014 Meldola, Italy; michele.zanoni@irst.emr.it (M.Z.); chiara.arienti@irst.emr.it (C.A.); anna.tesei@irst.emr.it (A.T.); 4Advanced Research Center on Electronic Systems (ARCES) for Information and Communication Technologies “E. De Castro”, University of Bologna, I-40125 Bologna, Italy; 5Department of Computer Science and Engineering (DISI), University of Bologna, I-40136 Bologna, Italy

**Keywords:** distribution density maps, subcellular mapping, data visualization, microscopy, fluorescence, image processing, computer-assisted, tumor cells, cultured

## Abstract

Subcellular spatial location is an essential descriptor of molecules biological function. Presently, super-resolution microscopy techniques enable quantification of subcellular objects distribution in fluorescence images, but they rely on instrumentation, tools and expertise not constituting a default for most of laboratories. We propose a method that allows resolving subcellular structures location by reinforcing each single pixel position with the information from surroundings. Although designed for entry-level laboratory equipment with common resolution powers, our method is independent from imaging device resolution, and thus can benefit also super-resolution microscopy. The approach permits to generate density distribution maps (DDMs) informative of both objects’ absolute location and self-relative displacement, thus practically reducing location uncertainty and increasing the accuracy of signal mapping. This work proves the capability of the DDMs to: (a) improve the informativeness of spatial distributions; (b) empower subcellular molecules distributions analysis; (c) extend their applicability beyond mere spatial object mapping. Finally, the possibility of enhancing or even disclosing latent distributions can concretely speed-up routine, large-scale and follow-up experiments, besides representing a benefit for all spatial distribution studies, independently of the image acquisition resolution. DDMaker, a Software endowed with a user-friendly Graphical User Interface (GUI), is also provided to support users in DDMs creation.

## 1. Introduction

Living cells are functionally defined by their anisotropy, as they rely on molecules distribution and compartmentalization to efficiently perform and control the biochemical reactions necessary for their life. Accordingly, abundance and especially molecules subcellular location are essential descriptors of their own behavior and function [1]. In fact, subcellular mislocation of many proteins [2,3] and RNAs classes [4,5] is associated to a variety of diseases, including cancer. Although most of studies focus on proteins, any other kind of targetable molecule can be virtually addressed, such as specific drugs [6], organelles markers [7] or other bioactive species [8]. These studies commonly rely on fluorescence imaging techniques for selective molecules visualization [2,3,5,6,7,8,9], but too often they limit themselves at visual annotation of targets location [10,11,12,13], which is mainly qualitative, subjective and prone to bias investigation [14].

Precise quantification of subcellular distribution of fluorescently marked molecules can be achieved by single-molecule localization microscopy (SMLM) techniques [15,16], which super-resolve via software the image beyond the physical limitation of optical resolution [17,18,19,20,21]. Despite its relevance, SMLM for routine and large-scale experiments is still restricted to large laboratories, due to technical and technological burdens. In fact, besides the high costs, the complexity of system calibration, image acquisition and computational reconstruction tasks [22,23] preclude SMLM technologies to most users [24] and thwart their adoption rate [25].

On the other hand, image-based techniques with proper algorithms can often compensate for insufficient instrumentations. In fact, image processing has become an integral tool in the daily activity of most laboratories, where it mainly serves the automation of procedures that have been manual for many years, thus providing fast, quantitative and repeatable measurements of imaged structures descriptors, such as object’s dimension and shape [26]. As a main example, preclinical in vitro studies, which typically assay drug efficacy and effectiveness by mean of commercial kits or user-validated protocols, have been profitably integrated with microscopy imaging as a further tool to complete the preclinical evaluation of antitumoral effects of the treatment investigated [27,28].

Regarding spatial distributions, image analysis permits to capture phenomena hidden from visual inspection [29], to measure molecules dispersion and its variations [30], and to construct 2D and 3D pseudo-color quantitative maps [31], which graphically describe the observed patterns in a highly intuitive and interpretable manner [2]. Even when working at non-super resolution, information conveyed by subcellular distribution can be preserved through image processing. As a main example, local image analysis permits to report pixel saliency as a function not only of the pixel itself, but also of its surroundings [32], thus permitting to reconstruct local geometric interactions, neither quantifiable, not even perceivable, in the native intensity images, which require proper post-processing. To this purpose, investigating pixel connectivity is a simple yet powerful tool to go beyond intensity analyses and access local image structures [33], as showed by popular applications as semantic segmentation [34] and image encoding and encrypting [35]. However, it may happen that after image pre-processing and thresholding the structural information of the “objects” of interest can be partly lost, yet more in case of structures whose size is close to system resolution.

In this work, we relax the connectivity constraint, introducing the novel concept of *local density* to semantically weigh the reference pixel by its neighboring information. This approach is quite robust against different resolutions, besides permitting to keep information also when classical connectivity analysis fails. This new concept also conveys three main innovations. The first is the local density index (LDI), representing pixel local density as the number of pixels, normalized by a unit area. LDI brings new knowledge regarding objects density, integrating distributions analysis with novel information. The second consists in the intrinsic property of local analysis to preserve the informativeness of spatial distributions, even in images with a reduced resolution. Third, the application of the method enables the simple and fast creation of density distribution maps (DDMs). Besides quantitatively describing the objects absolute and self-relative location, DDMs offer a pictorial view of the density distribution of the whole imaged sample, representing an unprecedented support to detect possible subpopulations.

The effectiveness of DDMs is exemplified through their application to four image sets acquired with confocal fluorescence microscopy and micro computed tomography (CT), which demonstrates how local neighbor analysis can enhance information confidence even in the absence of high-resolution technologies, and how DDMs extend their applicability outside the mere spatial distribution analysis. Efficiency and simplicity in building DDMs also permit continuous assessment of marker distribution in fluorescence images, thus enabling on-going monitoring and adjustment of experimental conditions. In conclusion, DDMs can concretely improve the outcome of experiments, independently of the resolution of the acquisition system.

Finally, we provide a software program, DDMaker, endowed with a user-friendly GUI, to support researchers in building the DDMs for their own experiments.

## 2. Materials and Methods

Five datasets are considered in this study: COL7, MG-63, A549 sh/p53, HeLa, micro-CT. As the first step, we aim at showing the greater informative content conveyed by the DDM in structure identification and how it changes with their application to the same dataset sampled at a different resolution. To this purpose, the images of COL7 are downsampled, thus deriving a sixth dataset at a halved resolution (COL7-h). In this way, COL7 can be used as the benchmark for the analysis performed with COL7-h. Secondly, we apply DDMs at three different subcellular distribution studies in MG-63, A549 sh/p53 and HeLa cells, imaged in homonymous datasets. Finally, we show the usefulness of DDMs in a mere technical context, to segment mice bone structures from micro-CT images.

Images of COL7 were downloaded free of charge from the public repository “The Image Cell Library” (CIL-CCDB) (http://www.cellimagelibrary.org) [36]. Images of A549 sh/p53 and HeLa cells were acquired for another study we are conducting to assess the effect of different stress conditions on RNA:DNA hybrids subcellular distribution, and are not public. The images of MG-63 and micro-CT were provided by the laboratories mentioned in the Acknowledgment Section and are not public.

### 2.1. Cell Culture and Image Acquisition

(1) Images of monkey kidney fibroblast COL7 cells (The Cell Image Library [36], CIL:13701) were acquired with a confocal fluorescence laser scanning microscope LSM 510 or LSM 710 (Carl Zeiss, Inc., Oberkochen, Germany, equipped with a Plan-Apochromat 63×/NA 1.4 objective in multitrack mode. COL7 cells were indirectly immunolabeled against the wild-type and the W164S mutant of the vasopressin V2 receptor (V2R) with anti-myc antibody (Ab), and against endosomes with anti–transferrin Ab [37]. (2) Images of human osteosarcoma MG-63 cells, exposed or not to paclitaxel-loaded nanoparticles (PTX-Ce6@ker_ag_), were acquired with a confocal fluorescence laser scanning microscope Ti-E A1R (Nikon, Amsterdam, The Netherlands) equipped with a 60×/NA 1.4 oil Plan-Fluo. MG-63 cells were indirectly immunostained against MTs with anti-β-tubulin Ab and incubated with Phalloidin-FITC for actin staining and with Hoechst for nuclei staining [38]. (3) Human lung adenocarcinoma A549 cells (ATCC, Manassas, VA, USA) were cultured in F12K (ATCC) supplemented with 10% FBS (Euroclone, Milan, Italy), 1% penicillin/streptomycin (GE Healthcare, Milan, Italy) and 2% amphotericin B (Euroclone, Milan, Italy), then plated at the density of 30,000 cells/well and infected with lentivirus LV-THM-sh-p53 at MOI = 10 TU/cell, as previously described [39]. Cells were seeded on a glass coverslip at the density of 30,000 cells/slide and underwent one-fraction 2-Gy gamma irradiation [40]. After 72 h cells were fixed and permeabilized with ice-cold methanol for 10 min and acetone for 1 min on ice, blocked with 2% BSA, stained with 1 µg/mL 4′,6-diamidino-2-phenylindole (DAPI) and immunostained for RNA:DNA hybrids (primary anti-S9.6 Ab (1:100 dilution, Kerafast, Boston, MA, USA), secondary goat anti-mouse Alexa Fluor 568 (1:250; Life Technologies, Carlsbad, CA, USA)). A549 sh/p53 cells were imaged with inverted confocal laser scanning microscope Eclipse Ti (Nikon Corporation, Tokyo, Japan) equipped with NIS-Elements Ar software. 12-bit images were acquired with a Plan Apo 60×/1.4 oil objective with lateral resolution of 0.1 µm/pixels and axial resolution of 0.2 µm/pixels. (4) Human cervix adenocarcinoma HeLa cells (ATCC, Manassas, VA, USA) were cultured in EMEM (ATCC) supplemented with 10% FBS (Euroclone, Milan, Italy), 1% penicillin/streptomycin (GE Healthcare, Milan, Italy) and 2% amphotericin B (Euroclone, Milano, Italy). Hyperbaric Oxygen Therapy (HBOT) was applied at 1.9 absolute atmosphere (ATA) in a hyperbaric chamber for 1 h. After 72 h, cells were fixed and permeabilized with ice-cold methanol for 10 min and acetone for 1 min on ice, blocked with 2% BSA, stained with 1 µg/mL 4′,6-diamidino-2-phenylindole (DAPI) and immunostained for RNA:DNA hybrids (primary anti-S9.6 Ab (1:100 dilution, Kerafast, Boston, MA, USA), secondary goat anti-mouse Alexa Fluor 568 (1:250; Life Technologies, Carlsbad, CA, USA)). HeLa cells were imaged with inverted confocal laser scanning microscope Eclipse Ti2-e (Nikon Corporation, Tokyo, Japan) equipped with NIS-Elements Ar software. 12-bit images were acquired with a Plan Apo 60×/1.4 oil objective with lateral and axial resolution of 0.1 µm/pixels. (5) Images of harvested mice tibiae were scanned with a micro-CT using a microfocus X-ray tube KEVEX PXS10-65W (Thermo Scientific Co., Waltham, MA, USA; 70 kV, 0.035 mA) and captured with a VHR1:1 CCD camera (Photonic Science Ltd., East Sussex, UK; 4008 × 2672 pixels, 9 μm pixel size). Final voxel size (2× magnification) was isometric 4.5 μm^3^.

### 2.2. Image Segmentation

All the image processing procedures are implemented in MATLAB^®^ (R2019a v.9.6.0, The MathWorks, Natick, MA, USA). (1) Marked V2R structures in COL7 and COL7-h cells are first segmented by grey level top hat filtering with disk-shaped structuring elements (SE) of fixed size (in μm), then thresholded at the 95th percentile. (2) MTs in MG-63 cells are segmented by ISODATA thresholding of single optical sections. (3) RNA:DNA hybrids in A549 sh/p53 cells are segmented as follows: (i) MIPs denoising by 8-bit quantization, (ii) nuclear region delineation by maximum entropy thresholding of DAPI signal, (iii) grey level top-hat enhancement with disk-shaped SE of fixed size (in μm) in cytoplasm, (iv) ISODATA thresholding in cell nucleus and cytoplasm, separately. (4) RNA:DNA hybrids in HeLa cells were segmented by ISODATA thresholding of MIPs’ positive values. (5) Tibial metaphyseal trabeculae in micro-CT images are segmented by (i) first performing an image denoising through 8-bit conversion, (ii) followed by an image contrast adjustment by adaptive histogram equalization, (iii) then thresholding the local intensity peaks by top-hat, and (iv) finally by applying our method to retain only densely distributed peaks, corresponding to metaphyseal trabeculae.

### 2.3. Local Distribution Analysis, LDI and DDM

DDMs creation is a two-step procedure (Figure 1).

First, a 2D-image (input) is segmented to achieve the foreground (binary) mask of the object(s) of interest (Figure 1a, left and center). Second, local distribution analysis (Figure 1b) is performed on the binary mask by assigning to each foreground pixel a value corresponding to the number of the foreground pixels in its neighborhood, defined by a rectangular (2n + 1) × (2m + 1) search window, with n and m being the half-sides along *X* and *Y* directions, respectively. As an example, without losing generality, in case of a 3 × 3 window, possible values for the reference pixel range from 0 for isolated pixels, to 8 for full-connected ones. In practice, the value assigned to each pixel represents its LDI. The image containing the set of LDI is a DDM, which can also be visually represented in pseudo-colors. Therefore, each DDM’s pixel is suggestive of the amount of information in its own neighborhood. It is worth noting that the method can be applied to any distribution study, only requiring a binary input image, independently of the imaging technique and the acquisition system resolution.

The most important implication of DDM can be seen when created with a 5 × 5 (or larger) search window, as illustrated in Figure 2.

Starting from a grey level input image from COL7 dataset and after an independent binarization procedure, the DDM is created (Figure 2a). Figure 2b shows how differently single and isolated pixels are semantically treated. Here, red isolated pixels are those having LDI = 0, while green isolated pixels have LDI = 2 or greater, as well as the white ones. Finally, purple pixels, with LDI = 1 may be either isolated or “end-point” towards no-density space. Isolated red pixels have no connections with any object in their 5 × 5 neighbors and are the best candidate to be removed since they do not apparently retain any information. On the contrary, it can be seen that green pixels are isolated, but not alone, suggestive to belong to structured though fragmented objects that should hence be kept. This is an example of semantic membership assignment of pixels based on their neighbor’s information. As LDI is function of single pixels, the objects can be composed of pixels with different LDI = 1, as it happens for all the white pixels. However, with a 5 × 5 window, aggregates of two pixels can be subject to an uneven behavior. In fact, depending on whether the aggregate is nearby a structured object or not, they can have one or both pixels with LDI = 1, respectively. Nevertheless, if undesirable this behavior can be modified with ad-hoc assumptions.

### 2.4. DDM’s Search Windows for the Analyzed Datasets

For COL7 dataset, a 5 × 5 search window is chosen for local distribution analysis in the imaged cells, this also permitting to have in COL7-h a halved-size search window (3 × 3) to perform the same analysis. For MG-63, A549 sh/p53 and HeLa datasets, a 3 × 3 search window is employed for local distribution analysis, since the objects of interest are in the range of few pixels and the smallest window is suited for detection and discrimination of single particles from small aggregation events. For micro-CT images, a search window of 29 × 159 pixels, approximating the real size of the imaged tibial metaphysis [41], is selected.

### 2.5. Assessment of Results

Identifying objects for either object counting, or to know their position, represents one of the most grounding steps in biological quantitative imaging. Therefore, we choose object counting to evaluate the ability of our method to identify single structures and, consequently, the effectiveness in characterizing their spatial distribution. Counting is carried out in COL7 segmented images, after a preliminary step needed to remove foreground pixels expectedly due to “noise” arising from sample preparation and/or acquisition process. Commonly, the denoising methods rely on area-based or connectivity-based thresholding. While the former aims at *removing* too small (or too big) aggregates, the latter also encodes neighboring information aiming at *preserving* connected pixels. Differently, we use our DDMs for a *density-based* thresholding and compare our counting with those achieved by area-based and connectivity-based thresholding that is, respectively, after removing 1-pixel (i.e., isolated) objects, or keeping pixels with 4-, diagonal- or full-connectivity by sequential image opening and closing with same 3 × 3 SE. From here on, objects are defined as 8-connected. The outcome is assessed through statistical measures derived from the contingency table (see Appendix B for details). In addition, to assess the robustness of the DDM to varying image quality, we compare the informativeness conveyed by DDMs applied to COL7 and to COL7-h, having a halved resolution, by performing a pattern matching using the normalized cross correlation (NCC).

The MG-63, A549 sh/p53 and HeLa datasets are used to exemplify the different benefits of applying DDMs in subcellular distribution analysis, aiming at identifying a discriminant feature (descriptor) of different cell conditions. A visual inspection of DDMs in all datasets suggests us that the LDI percentage (i.e., the ratio between the number of pixels with given LDI and the number of all analyzed pixels) could be a suitable descriptor. Nevertheless, in the HeLa dataset, the number of objects (or better, blobs) composing each density level (weighted by cell area) was also considered, in order to refine the assessment of the spatial gathering of pixels sharing a same LDI.

Finally, the micro-CT dataset is used to exemplify the applicability of the method beyond the pure molecular distribution analysis, by using DDM to integrate the image segmentation procedure of tibial metaphyseal trabeculae.

### 2.6. Statistical Analysis

Statistical analyses are performed in MATLAB^®^. Data deviation from normality is early verified by histogram inspection, followed by the Shapiro-Wilk test, based on which the discriminatory power of descriptors is assessed by either two-tail Student’s *t*-test or Wilcoxon rank-sum test. *p*-values < 0.05 were considered for statistical significance.

## 3. Results and Discussion

### 3.1. DDMs Are Effective and Robust to Quantify Spatial Distributions

The DDM’s capability to increase confidence in spatial distribution measurement is assessed by its application to object counting, in comparison with area- and connectivity-based thresholding (Figure 3a).

Specifically, starting from a binarized COL7 image (Im): (1) DDM is created and binarized after that isolated pixels (LDI = 0) are removed (DDMm); (2) area-based denoising is performed by removing isolated pixels (Im1); (3) 4- (Im2), diagonal- (Im3) and full- (Im4) connectivity-based denoising are carried out by morphological opening and closing (Figure 3a, the squares with the reference central red pixel).

As reported in Table 1, DDMm is the image that better approximates the object counting in the original binarized image Im. Indeed, it shows by far the highest TPR = 63%, and the lowest FNR = 37%, accordingly. The non-complementarity of TP and FN in Im2, Im3 and Im4 hints at a fragmentation of Im objects induced by the connectivity-based denoising, as suggested by the 24%, 18% and 22% of objects that are detected with an incorrect (i.e., ≥1:2) stoichiometric detection rate (SDR) in these images, respectively (see Appendix B for details). On the contrary, neither the area-based nor our method fragment the objects. Finally, DDMm shows the highest Overlap (i.e., the best match) with Im, this suggesting a better accuracy in estimating position and object extension. It is worth noting that the overlap difference between DDMm and Im1 is attributable to the 1-pixel objects that, being isolated but not alone, are discarded in the latter, but not in the former.

Finally, we used the NCC to perform a pattern matching between each analyzed image and its half-resolved counterpart (Figure 3b and Table 2). While the best result is in the presence of downsampling only (NCC = 0.96, that is, 4% information loss), DDMm retains the highest correlation between results achieved at full and half resolution (NCC = 0.79), meaning that it shows the highest robustness against resolution reduction.

For all these reasons, DDMm represents the best option for object detection, and consequently for distribution analysis, as it minimizes FN, maximizes TP and the object detection with correct stoichiometry. Therefore, our method can be preliminary considered also as an effective denoising procedure in itself, that besides retaining objects on the basis of their *connectivity*, can even keep the most informative ones on the grounds of their *local density*. More importantly, this reinforcement of each pixel position by exploiting information from surroundings, makes our method to be the most robust to resolution variation. This means that, independently of the resolution of the acquisition device, our method can effectively improve the informativeness of the distribution analysis.

### 3.2. DDMs Disclose Hidden Distribution Properties

This example shows how to use local density information to strengthen ordinary analyses. Figure 4 addresses the MTs resolving in confocal images of MG-63 cells exposed to Paclitaxel-loaded nanoparticles (PTX-Ce6@kerag) (Figure 4a) [38].

As Paclitaxel (PTX) is expected to suppress MTs dynamic instability [42], the MTs signal is investigated through the optical sections. For visualization purposes, confocal sections are summarized in MIPs (Figure 4a, top color images). By comparing untreated control (left) and PTX-treated (right) cells, a different subcellular location and intensity of MTs (red) can be noticed. This visual consideration still holds for single optical sections (grey level images, bottom left) and it is supported by MTs intensity quantification (Figure 4b: median intensity in treated cells greater than 126%, *p* < 10^−5^, Appendix A. Coefficient of variation (CV) through sections: 0.55 (CTR), 0.44 (PTX)). However, local density analysis of MTs in single sections discloses a hidden aspect of the distribution. Indeed, the line plots of LDI percentage in CTR (Figure 4c) show a marked presence of the highest LDI = 8 and a reduced presence of LDI between 0 and 7, which are also more stable through the optical sections (average CV: 0.70). After PTX delivery (Figure 4d), the presence of all LDI becomes constant through sections (average CV: 0.15) and, most important, LDI = 8 becomes nearly exclusive and the remaining LDIs almost disappear, since the former significantly increases (+69%, *p* < 10^−5^), while the latter decrease (−68% on average, always *p* < 10^−3^, Appendix A). The predominant LDI = 8 presence could be ascribed to the dense and crystallized MTs appearance induced by high PTX concentration [43]. Together with LDI = 8 constant presence throughout optical sections, this finding suggests that Ce6@kerag-mediated PTX delivery is probably even more efficient than what reported by the authors themselves, hence highlighting the prominence of such a delivery system for clinical application.

### 3.3. DDMs Can Capture Relevant Spatial Distributions Blind to Visual Inspection

This example is probably the most effective one to show how the hidden information disclosed and quantified by DDMs can provide added knowledge. Figure 5 reports A549 sh/p53 cells untreated (CTR) or subject to gamma irradiation (2 Gy), marked against RNA:DNA hybrids to assess how their subcellular distribution varies in response to treatment.

An earlier image comparison between irradiated and non-irradiated cells (Figure 5a, left) suggests that hybrids differently redistribute in cytoplasm and nucleus after cell irradiation, displaying the emptying of nucleus and a pan-cytoplasmic dispersion of hybrids. However, DDMs computation (Figure 5a, center) unveils that what appeared as an uninteresting cytoplasmic redistribution unexpectedly consists of an accumulation of hybrids in what looks like a cytoplasmic perinuclear ring, that after nuclear boundary segmentation results to lie inside (Figure 5a, magnification, right). DDMs analysis allows quantifying a significant increase in both cell nucleus and cytoplasm of low LDI percentages (LDI = {0,1,2,3}, *p* < 10^−4^), coupled with a symmetric decrease in high LDI percentages (LDI = {5,7}, *p* < 0.006 in cytoplasm and LDI = 8, *p* = 0.012 in nucleus) (Appendix A). This means that 2-Gy irradiation leads to a hybrids de-condensation in both compartments, more heavily in cell nucleus, where the decrease involves higher density levels. Although this evidence would seem to disagree with the clear perinuclear hybrids crowding at 2 Gy, the de-condensation regards the whole cellular compartments, while the hybrid accumulation occurs at the sub-regional level. Notably, we can conclude that, despite the significant changes in LDI percentages, a 2 Gy irradiation can be said to peculiarly affects hybrids subcellular location, rather than aggregation state and density, accordingly. This finding highlights the need of local processing and the importance of DDMs to convey both quantitative and visual information, which have to be considered together to assist researchers in capturing the complexity of phenomena.

### 3.4. DDMs Can Detect and Quantify Sample Heterogeneity

This example shows how DDMs can be used to disclose and discriminate subsamples by the local density distribution of marked structures. Figure 6 reports HeLa cells exposed (HBO 1.9 ATA) or not (CTR) to hyperbaric oxygen conditions and marked against RNA:DNA hybrids to assess their subcellular distribution variation in response to such stressing condition.

Visual inspection of acquired images (Figure 6a) suggests a difference in hybrids signal intensity and distribution between treated and untreated cells. This difference is confirmed and stressed by DDMs, which moreover disclose a heterogeneous hybrids subcellular distribution among HBO-treated cells, identifying three cell subgroups characterized by a cortical, scattered, and intermediate distribution, respectively (Figure 6a, colored annotations). DDMs creation permits to differentiate the three distributions by both LDI percentage and number of blobs (Figure 6b, Appendix A), where the latter varies more than the former, meaning that the groups are not so much characterized by different densities as they differ in the way the densities are spatially distributed. When grouping all HBO-treated cells together, this heterogeneity results in a higher variance in spite of the increased number of samples (Appendix A), with consequent weakening of statistical comparison between treated and control group [44]. In conclusion, in this case DDMs provide some unprecedent information. First, they indicate that the sample may be not large enough to account for the heterogeneity of the entire population, and that a careful outliers detection and removal is needed before data analysis. Second, DDMs reveal that cells of a same subgroup are spatially gathered, thus raising doubts on the homogeneity of the created hyperoxic environment and suggesting new experiments, under strictly controlled conditions, also aiming at investigating the dependence of hybrids distribution on the oxygen concentration. At the end, independently of the sample heterogeneity, DDMs already reveal that hyperbaric conditions induce a redistribution of hybrids and a change in their condensation state.

### 3.5. DDMs Can Apply Beyond Distribution Analysis

In the previous paragraphs, we showed how DDMs can reinforce, supplement or disclose distribution information. However, applicability of DDMs extends beyond distribution analysis in microscopy cell imaging, for instance, to improve image segmentation procedures. Figure 7 describes the main steps of the automated segmentation of metaphyseal trabeculae from a Mus musculus tibiae in micro-CT images (Figure 7a), that involves local density analysis.

First, the image contrast is enhanced by 8-bit conversion (Figure 7b) and adaptive histogram equalization (Figure 7c). Then, a top hat filtering of the image (Figure 7d) with proper SE (i.e., with dimension comparable to that of trabeculae to be segmented) permits to retain local intensity peaks (corresponding to more mineralized structures), while disregarding irrelevant pixels with low values (corresponding to bone cavities). As this procedure well identifies metaphyseal trabeculae, it also includes unwanted information from other bone structures. To isolate metaphyseal trabeculae, local density analysis can be used with denoising purpose, when selecting an appropriate local window size (i.e., with dimension comparable to that of metaphysis to be segmented). This way, the resulting DDM (Figure 7e) permits to distinguish metaphysis trabeculae as the denser mineralized structures, and to exploit this information to segment them (Figure 7f).

### 3.6. GUI for DDMs Creation

To allow users, even with basic skills, to build DDMs we supply DDMaker (Figure A1), a software program endowed with a user-friendly GUI, created with MATLAB^®^ App Designer, which does not require any training or expertise before using. In few steps, the software permits to customize the search window size and to create the DDMs either directly, starting from binary images, or indirectly, from RGB or grey level images, thanks to a dedicated module for adaptive image binarization. Moreover, DDMaker allows visualizing the DDM-derived colormaps and to perform an original DDM’s density-based thresholding, useful for the automated image segmentation. The software builds DDMs and save all data from few seconds to minutes on entry-level computers (e.g., a dataset of one hundred grey level images is fully processed in a little more than one minute on a PC endowed with Intel i3-4005U, 1.70 GHz processor, and 8 GB RAM). The simplicity in creating and interpreting DDMs, jointly with their effectiveness, make DDMaker a valuable tool for fast assessment of target distributions. All considered, DDMs could serve as a crucial check-point for long-lasting experiments, as well as for follow-up and large-scale studies, that can be monitored on-line and corrected in progress, or even stopped, based on the continuous feedback by DDMs. It is worth noting that this allows optimizing time and costs by adjusting or rapidly restarting experiments that would otherwise have been discarded, just after ending. A detailed description of functionalities and tasks of DDMaker is provided in Appendix C. DDMaker is available as a public open-source software written in MATLAB^®^ and as a 64-bit stand-alone application (https://sourceforge.net/projects/DDMaker).

## 4. Conclusions

In this paper, we introduce an innovative method for subcellular distribution analysis, able to semantically quantify the local density of pixels, summarized as the Local Density Index, finally exploited to build a Density Distribution Map in pseudo-colors to prompt visual survey of the distributions. Using DDMs lead to a more accurate estimation of molecules position, and increased robustness to resolution variations, if compared with the standard approaches. This allows DDMs to characterize and quantify both evident and hidden subcellular distribution, thus opening to the formulation of new biological considerations. As such, DDMs appear as an innovative tool to supplement intensity analysis even for visual assessment, besides quantification of signal distribution. DDMs can also be integrated in a standard image processing pipeline. In fact, the method shows its effectiveness to perform a smart denoising, which selectively addresses single pixels based on their neighborhood’s structural information. In addition, our method can be used for density-driven segmentation, which allows a good identification of small and thin morphological structures, like in the micro-CT images, that otherwise would have been merged. Finally, it is worth noting that as a resolution-independent technique enhancing the detection of native information DDMs can also benefit high-resolution technologies. DDMs computation is within every user’s reach with the DDMaker software we provide. The immediacy of DDMs creation, besides the exemplifying applications herein considered, allows DDMs to be employed in continuous monitoring routine and large-scale experiments, planning and progression of explorative investigations as for example in the study of cancer cell biology. In particular, DDMs analysis permits to detect heterogeneous responses to treatment in cell sub-populations, improving clinical drug development and with the potential to impact decisively on medicine in general and on oncology in particular.

As regards the limitations, the first is that DDMs can be applied to binary images only, although this is intrinsically due to the design of the method itself. The second limitation is that, for this reason, DDMs require that previous image acquisition and segmentation steps have been properly carried on. For this reason, DDMaker is also endowed with a segmentation module.

## Figures and Tables

**Figure 1 sensors-21-01009-f001:**
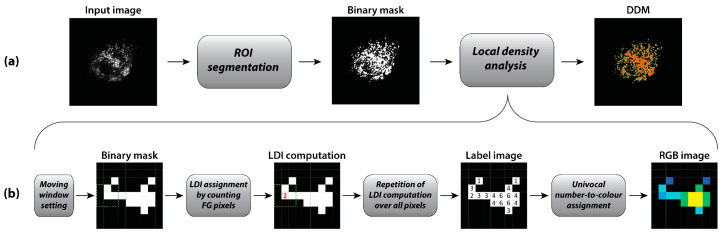
Flowchart of a DDM creation pipeline. (**a**) The acquired image is segmented in a binary mask. Then, the mask connectivity is explored by local density analysis to create the DDM in pseudo-colors. (**b**) Details of local density analysis: after setting the search (moving) window size, each foreground (FG) pixel of the binary mask is assigned a number representing the amount of FG pixels in its locality (i.e., LDI), this constituting the input to build the pseudo-color DDM.

**Figure 2 sensors-21-01009-f002:**
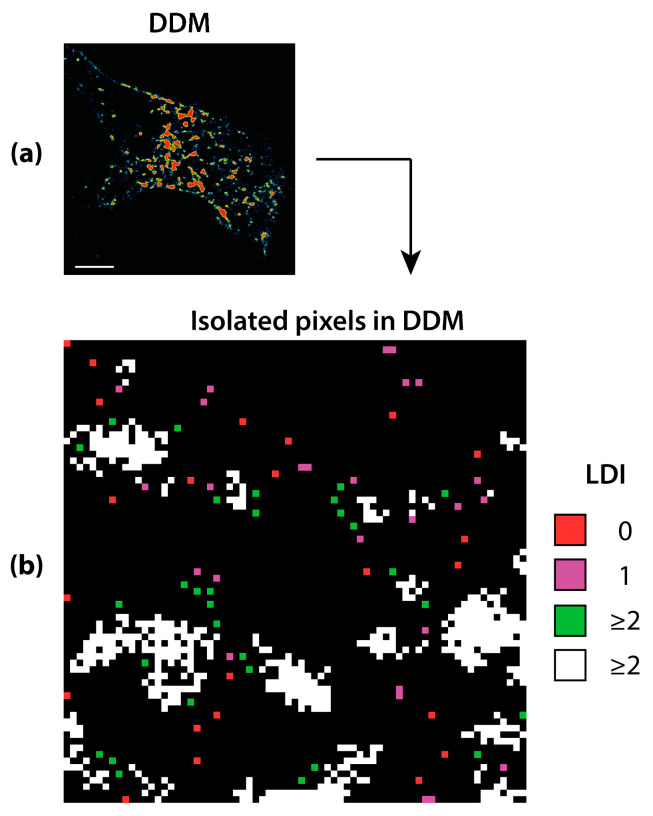
DDM mask creation. The acquired COL7 image is firstly segmented and the DDM (**a**) is then created with a 5 × 5 search window, that is the minimal window size that allows discriminating single pixels based on their locality (**b**) Pixels semantics: (1) isolated red pixels, with LDI = 0; (2) green pixels, isolated, but not alone in their 5 × 5 neighborhood, with LDI ≥ 2; (3) either isolated or “end-point” purple pixels, with LDI = 1; (4) connected white pixels, with LDI ≥ 2. Scale bar: 5 µm.

**Figure 3 sensors-21-01009-f003:**
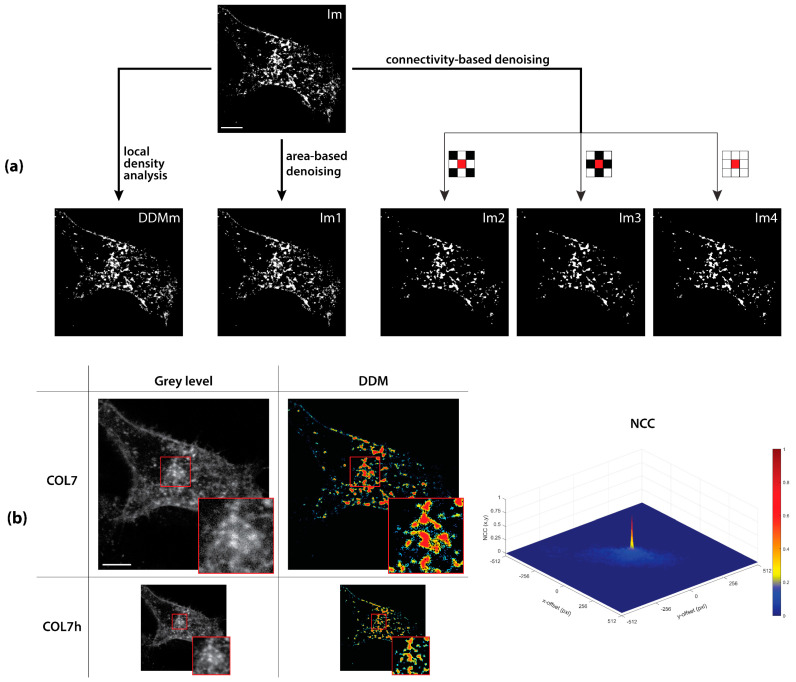
DDMs are effective and robust to quantify spatial distributions. (**a**) After signal binarize Table 1. removed), area- (Im1, 1-pixel object removal) or connectivity-based denoising (Im2, Im3 and Im4, 4-, diagonal- and full- connectivity-based denoising, respectively). (**b**) The signal accuracy reduction caused by resolution halving of COL7 to COL7-h is better resisted by our method application, as quantified by a maximum NCC coefficient of 0.79 for COL7 and COL7-h DDMs. The colorbar indicates the normalized function values. Scale bars: 5 µm.

**Figure 4 sensors-21-01009-f004:**
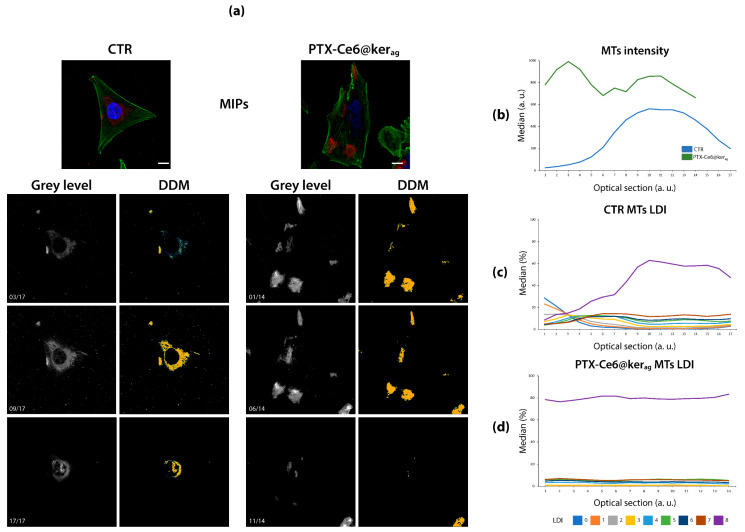
DDMs disclose hidden distribution properties. (**a**) Top: RGB MIPs of MG-63 untreated (left) or exposed to PTX-Ce6@kerag nanoparticles (right) cells, stained for DNA (blue) and actin (green), and immunolabeled against and β-tubulin (red). Bottom: exemplificative optical sections (left) and DDMs (right) of β-tubulin signal distribution along *Z*-axis. Line plots of MTs median intensity (**b**) and percentage LDI distribution in CTR (**c**) and PTX-treated cells (**d**) along the *Z*-axis. In PTX presence, MTs are brighter (+126% on average, *p* < 10^−5^), denser (+69%, on average, *p* < 10^−5^) and more present at a high density through the sections (average CV through sections for LDI = 8:0.02). Scale bars: 10 µm.

**Figure 5 sensors-21-01009-f005:**
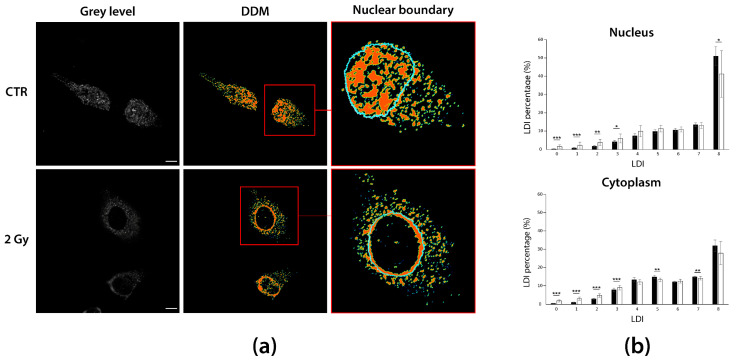
DDMs can capture relevant spatial distributions blind to visual inspection. (**a**) Grey level images of untreated and 2-Gy irradiated A549 sh/p53 cells are used to compute DDMs for immunostained RNA:DNA hybrids. DDMs highlight a perinuclear hybrids crowding inside the nucleus of 2-Gy irradiated cells. (**b**) Bar graphs of LDI percentages in the main cell compartments. A549 sh/p53 2-Gy irradiation induces hybrids de-condensation in both nucleus and cytoplasm, although with a slightly different magnitude. * *p* < 0.05; ** *p* < 0.01; *** *p* < 0.001. Scale bars: 10 µm.

**Figure 6 sensors-21-01009-f006:**
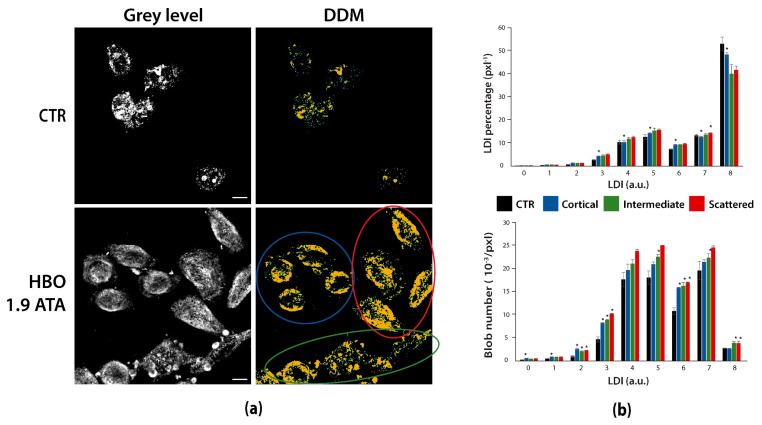
DDMs can detect and quantify sample heterogeneity. (**a**) Grey level images and DDMs of untreated (CTR) and 1.9 ATA HBO-treated HeLa cells marked against RNA:DNA hybrids. DDMs separate three cell groups of cortical (blue), scattered (red) and intermediate (green) hybrids distributions among HBO-treated cells. (**b**) Bar graphs of LDI percentages and derived blob number in HeLa cells. * *p* < 0.05 for statistical comparison of the three groups with the untreated control. Scale bars: 10 µm.

**Figure 7 sensors-21-01009-f007:**
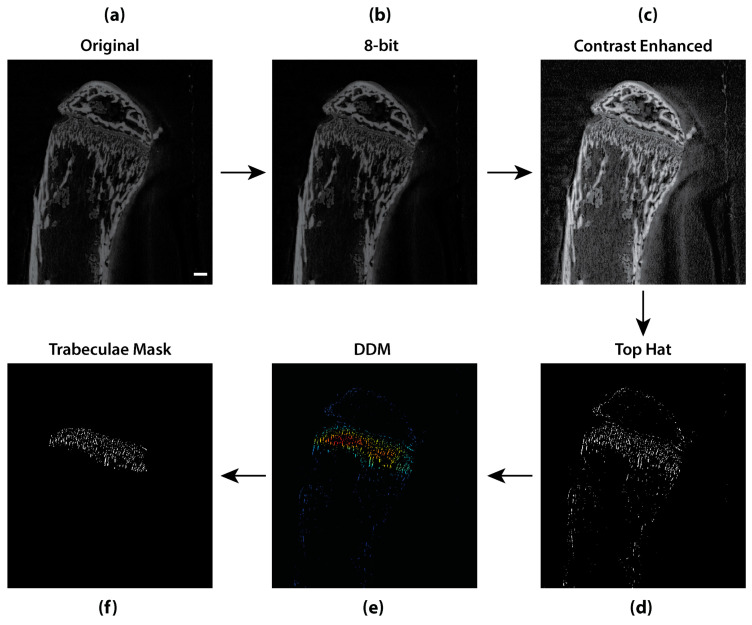
DDMs application to image segmentation. DDMs can applicate beyond distribution analysis in microscopy imaging: here, the segmentation of metaphyseal trabeculae in micro-CT (coronal) images of Mus musculus tibiae. The acquired image (**a**) is 8-bit converted for denoising purposes (**b**), contrast-enhanced by local adaptive histogram equalization (**c**) and top hat-filtered for the segmentation of local intensity peaks (**d**). This procedure well detects metaphyseal trabeculae, but also includes less dense signal from other bone structures. DDMs (**e**) permit to discriminate metaphyseal trabeculae as the denser local peaks in the image, and accordingly to segment them based on their local density (**f**). Scale bar: 200 µm.

**Table 1 sensors-21-01009-t001:** Contingency table for object counting with density-, area- and connectivity-based approach.

Image ^1^	TP	FN	TPR	FNR	SDR (%)		Overlap
			(%)	(%)	1:1	≥1:2	(%)
DDMm	1089	649	63	37	100	0	95
Im1	614	1124	35	65	100	0	92
Im2	170	1612	10	93	76	24	53
Im3	139	1625	8	93	82	18	51
Im4	94	1666	5	96	78	22	42

^1^ Ground truth Im No. of objects = 1738. Abbreviation list: Appendix A.

**Table 2 sensors-21-01009-t002:** Comparison between COL7 and COL7-h.

Image ^1^	NCC
Im	0.96
DDMm	0.79
Im1	0.75
Im2	0.73
Im3	0.75
Im4	0.71

^1^ Abbreviation list: Appendix A.

## Data Availability

No new data were created or analyzed in this study. Data sharing is not applicable to this article.

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
