# Peer review of "Density Distribution Maps: A Novel Tool for Subcellular Distribution Analysis and Quantitative Biomedical Imaging"

_sensors, 2021, doi:10.3390/s21031009_

Round 1
Reviewer 1 Report
This study highlights the usefulness of density distribution maps (DDMs) as a tool for subcellular distribution analysis and quantitative biomedical imaging through the use of a specific software called DDMaker. The manuscript includes seven keywords, seven figures with 20 sections, two tables, two appendixes with another two tables and one figure, seven supplementary tables, and forty-five references. Overall, it is a correct, very complete, and well-documented article, although some slight remarks are made.
The manuscript presents seven keywords. For keywords, where possible, please use Medical Subject Headings Terms (MeSH Terms). “Data visualization” is already a MeSH term. The following terms “distribution density maps”, “quantitative subcellular localization”, “local density”, and “local image analysis” are not MeSH terms. Some alternative MeSH terms proposed are “microscopy, fluorescence” better than “fluorescence microscopy”, and “cell line, tumor” rather than “cancer cell”.
Please provide more information about the MATLAB® program (version, company, etc.)
In Table 1 and 2 legends, please consider including the explanation of the abbreviations.
You could transfer the information from supplementary table S7 (Abbreviations) to table 1 and 2 legends.
References list
In some references, only the first author is cited followed by et al. This does not seem to agree with the journal's reference guidelines. You may consult them in the following link https://www.mdpi.com/files/authors/mdpi_references_guide.pdf
No further comments about the rest of the sections of the manuscript. All it's fine.
Reviewer 2 Report
The present study provides an image processing method for analyzing sub-cellular structure by comparing each pixel's position relative to the surrounding environment. The model seems novel and the presentation of results is comprehensive.
I would suggest reviewing the flow of the text and try to make it as simple as possible when necessary as it sometimes becomes difficult to follow. In addition, I would like to have your position on using this method for macro size objects: specifically organ and tissue analysis in surgical video processing.
Reviewer 3 Report
This paper proposes a method that allows resolving subcellular structures location by reinforcing each single pixel position with the information from surroundings. Although designed for entry-level laboratory equipment with common resolution powers, our method is independent from imaging device resolution, and thus can benefit also super-resolution microscopy. The approach permits to generate density distribution maps (DDMs) informative of both objects’ absolute location and self-relative displacement, thus practically reducing location uncertainty and increasing the accuracy of signal mapping. I read the manuscript with great interest and believe its topic is important and relevant. Although the manuscript is overall well-written and structured, it might benefit from additional spell/language checking.
What was the key motivation behind using Density distribution maps?
What are some key issues that present study has addressed?
Put the research limitations of the present study in the conclusion section.
Authors should further clarify and elaborate novelty in their contribution.
Besides do compare the present study findings with the past studies that used same datasets and highlight why your work is better. And in this context it is worth mentioning their experimental evaluation protocol for a fair comparison.
There are several interesting papers that look into image analysis. For instance, the below papers has some interesting implications that you could discuss in your introduction and how it relates to your work.
Chowdhary, Chiranji Lal, et al. "Analytical study of hybrid techniques for image encryption and decryption." Sensors 20.18 (2020): 5162.
Deng, Hanbing, et al. "Depth Density Achieves a Better Result for Semantic Segmentation with the Kinect System." Sensors 20.3 (2020): 812.
Round 2
Reviewer 3 Report
All my comments are addressed hence manuscript is accepted.